# Preparation and Properties of Natural Polysaccharide-Based Drug Delivery Nanoparticles

**DOI:** 10.3390/polym15112510

**Published:** 2023-05-30

**Authors:** Xuelian Chen, Lijia Liu, Chen Shen, Fangyan Liu, Enyu Xu, Yin Chen, Wang Jie

**Affiliations:** 1College of Food and Pharmacy, Zhejiang Ocean University, Zhoushan 316000, China; zhang17364621814@outlook.com (X.C.); fangyanliuliu@outlook.com (F.L.);; 2Key Laboratory of Experimental Marine Biology, Institute of Oceanology, Chinese Academy of Sciences, Qingdao 266071, China

**Keywords:** layer-by-layer, nanoparticle, polysaccharide

## Abstract

In recent years, natural polysaccharides have been widely used in the preparation of drug delivery systems. In this paper, novel polysaccharide-based nanoparticles were prepared by layer-by-layer assembly technology using silica as a template. The layers of nanoparticles were constructed based on the electrostatic interaction between a new pectin named NPGP and chitosan (CS). The targeting ability of nanoparticles was formed by grafting the RGD peptide, a tri-peptide motif containing arginine, glycine, and aspartic acid with high affinity to integrin receptors. The layer-by-layer assembly nanoparticles (RGD-(NPGP/CS)_3_NPGP) exhibited a high encapsulation efficiency (83.23 ± 6.12%), loading capacity (76.51 ± 1.24%), and pH-sensitive release property for doxorubicin. The RGD-(NPGP/CS)_3_NPGP nanoparticles showed better targeting to HCT-116 cells, the integrin αvβ3 high expression human colonic epithelial tumor cell line with higher uptake efficiency than MCF7 cells, the human breast carcinoma cell line with normal integrin expression. In vitro antitumor activity tests showed that the doxorubicin-loaded nanoparticles could effectively inhibit the proliferation of the HCT-116 cells. In conclusion, RGD-(NPGP/CS)_3_NPGP nanoparticles have potential as novel anticancer drug carriers because of their good targeting and drug-carrying activity.

## 1. Introduction

Currently, many new substances and nanotechnologies are being investigated rapidly to produce nanoparticles (NPs) for drug delivery and especially for cancer therapy [1,2]. NPs, as minute particles that are typically less than 200 nm in diameter, bring significant advances for medical purposes, such as improving the physicochemical properties of drugs, facilitating intracellular uptake, increasing targeting, and reducing their toxicity and side effects [3,4]. The attraction of NPs in disease treatment is based on their unique features, such as their relatively large functional surface and excellent ability to carry other compounds [5]. In detail, NPs have a good surface-to-mass ratio and can easily bind, encapsulate, and carry various drugs, probes, and biomacromolecules such as proteins. The functional surfaces enable tunable surface chemistry for multivalent surface modification with targeting ligands and then efficient identification of the complex in vivo environment [6]. Compared with most nanoparticles, the hollow core-shell structure of nanoparticles has unique properties such as uniform size, well-defined morphology, a large specific surface area, low density, and a high internal cavity, making them suitable for a wide range of potential applications [7,8]. Such applications make use of the large empty cavity in these hollow structures, which can be used for loading more drugs or active molecules. In addition, through the design and modification of the polymer cover of the nanoparticles, these drug loading systems can achieve a response to the microenvironment of the body and allow a targeted and controlled release, which is much superior to a conventionally uncontrolled release [9,10]. Thus, this kind of nanoparticle has a broader application prospect.

Apart from nanoprecipitation, emulsion diffusion, and double emulsification polymer coating methods, the preparation of nanoparticles has mostly used layer-by-layer (LbL) assembly technology in recent years [11,12,13]. This method needs a colloidal template on which a polymer layer is adsorbed either by incubation in the polymer solution and then washing or by decreasing the polymer solubility by the addition of a miscible solvent [14]. The commonly used ideal material of the polymeric layer of nanoparticles, such as chitosan, heparin, polylysine, gelatin, or dextran sulfate, may possess polyanionic or polycationic properties [15] with many advantages like biocompatibility, biodegradability, mucoadhesiveness, and permeability enhancer properties [16]. In the literature, a multilayer hollow nanocapsule with chitosan (CS) and carrageenan (Car) as the polycation and polyanion, respectively, has been fabricated via the sequential LbL electrostatic self-assembly method from the sacrificed template nanospheres (SiO_2_-NH_2_) with a diameter of 100 nm and responsiveness to pH and ionic strength [17,18].

Since there are many kinds of capsule wall materials used in the LbL assembly method, the selection of appropriate polymers as the materials can give the nanoparticles excellent performance on drug delivery. A new polysaccharide resource from *Nicandra physalodes* (Linn.) Gaertn. (NPGP) was recognized as a new pectic polysaccharide with a high content of galacturonic acid (85%) with the linear linkage of α-(1→4)-D-GalUA and a low methoxylation modification (15%) [19]. NPGP can form self-supporting gels with special properties. Low pH led to the suppression of electrostatic repulsions between the pectin chains, promoted chain-chain association mainly through hydrogen bonding, and formed a strong and stable gel. Thus, NPGP is a new gelling polysaccharide that shows great potential in nanocapsules [20,21,22]. In this paper, nanoparticles based on the novel polyanionic polysaccharides NPGP and CS were prepared by LbL technology using silica as a template. The layers of nanoparticles were constructed based on the electrostatic interaction between modified NPGP and chitosan (CS). Moreover, “targeted delivery” means a particular location inside a body or organ that is precisely targeted using a very specific drug delivery system. Therefore, a controlled drug delivery system combined with specific targeting capability is a very potent technology to be explored and exploited. Thus, the targeting ability of the nanoparticles was achieved by grafting L-arginine-L-glycine-L-aspartic acid (RGD) peptides that specifically bind to overexpressed integrin αvβ3 in tumor-associated neovasculature on NPGP. The drug loading properties and targeting of the nanoparticles (RGD-(NPGP/CS)_3_NPGP) were also researched.

## 2. Materials and Methods

### 2.1. Materials and Reagents

In our previous experiments, NPGP was extracted from *Nicandra Physalodes* (Linn.) Gaertn. seed with a molecular weight of 1.0 × 10^6^ Da and a pKa of 3.48. Chitosan (CS) with a molecular weight of 1.5 × 10^5^ Da, a pKa of 6.5, and a degree of deacetylation ≥ 90% from LMAl Bio Co. (Shanghai, China). Doxorubicin hydrochloride (DOX) from Aladdin Industrial Co. (Shanghai, China). The 3-(4,5-dimethtl-2-thiazolyl)-2,5-diphenyl-2H-tetrazolium bromide (MTT), high glucose DMEM medium, and Trypsin-EDTA are from Solarbio Science & Technology Co. (Beijing, China). Other reagents used were analytically pure from Sinopharm Chemical Reagent Co., Ltd. (Shanghai, China).

### 2.2. Preparation of SiO_2_-NH_2_ Nanospheres

In this study, nanosized silicon dioxide microspheres were prepared using the modified Stober method [23,24]. Specifically, 4 mL of tetraethoxysilane (TEOS) was added to a mixture of 100 mL of anhydrous ethanol and 6 mL of ammonium hydroxide under ultrasonic conditions. After ultrasonic treatment (ultrasonic power 120 w, frequency 53 Hz) for 2 h, the mixture was aged overnight to obtain a silicon dioxide-ethanol dispersion. Then, APTES (0.2 mL) was dropwise added to the dispersion slowly under high-speed stirring conditions, and the reaction lasted for 12 h [25]. After the reaction, the supernatant was removed by centrifugation (10,000 rpm/min, 10 min) and then washed with ethanol, an ethanol-water mixture, and deionized water in sequence until the solution was neutral to obtain amino-modified silica (SiO_2_-NH_2_).

### 2.3. Preparation of Polyelectrolyte Multilayer-Coated SiO_2_(NGPG/CS)_3_NPGP Nanoparticles by the LbL Method

The CS/NGPG multilayers were assembled on the SiO_2_-NH_2_ nanospheres by the LbL deposition technique [16,26]. SiO_2_-NH_2_ (0.5 g) was ultrasonically dispersed in 50 mL of 0.4% acetic acid solution (ultrasonic power 120 w, frequency 53 Hz, time 20 min), and then the dispersion was added to the pectin (1 mg/mL, 50 mL, pH 3.0) with magnetic stirring, followed by centrifugation of the nanospheres (10,000 rpm/min, 10 min) after stirring for 1 h. The excess polyanionic polysaccharide was removed by three cycles of centrifugation (10,000 rpm/min, 10 min) with pure water washing. As described above, the NPGP/CS bilayer construction was performed by dropwise adding the nanosphere dispersion solution to the CS solution (1 mg/mL in 0.4% (*v*/*v*) acetic acid solution). The alternate deposition of CS and NPGP was repeated until the deposition of 7 layers. The synthesized nanospheres were labeled as SiO_2_(NGPG/CS)_3_NPGP nanospheres.

### 2.4. Preparation of Polyelectrolyte Multilayer-Coated RGD-(NGPG/CS)_3_NPGP Nanoparticles

The SiO_2_(NGPG/CS)_3_NPGP nanospheres obtained in the previous step were ultrasonically dispersed in deionized water (8 mL), followed by the addition of 24 mL of 2 M HF/8 M NH4F buffer solution (1:2, *v*/*v*), stirred gently at room temperature to etch the SiO_2_-NH_2_ template [27]. Then, the mixture was dialyzed in water for 3 days to remove excess products such as NH4F. Finally, the (NGPG/CS)_3_NPGP nanoparticles were obtained through freeze-drying (prefreezing temperature −20 °C; time 12 h; freezing temperature −54 °C; vacuum pressure < 0.1 MPa).

Since the outermost layer of SiO_2_(NGPG/CS)_3_NPGP hollow nanoparticles is NPGP, the RGD peptide can be modified to the carboxyl groups of the outermost sugar chain by an amide reaction [28]. Briefly, 100 mg (NGPG/CS)_3_NPGP hollow nanoparticles were dissolved in DI water. The carboxyl groups of NPGP were activated by the addition of EDC (3 mmol) and NHS (3 mmol) for 2 h. After that, RGD peptide solution (1 mg/mL, 5 mL) was added and stirred for 24 h. The product was further purified by dialyzing for 24 h to remove unreacted reactants (MWCO = 3500). The samples were freeze-dried and stored at −20 °C.

### 2.5. The Characterization of NGs

The FT-IR spectra of SiO_2_, SiO_2_-NH_2_, NPGP, CS, and SiO_2_(NGPG/CS)_3_NPGP were measured from a frequency range of 4000 to 500 cm^−1^. Hydrodynamic diameter and surface charge of NGs were measured by a DLS instrument (Zetasizer Nano-ZS90) at 25 °C. The morphology of NGs was observed by TEM (JEOL JEM-2100F, Japan). The thermogravimetric analysis of SiO_2_(NGPG/Cs)_3_NPGP nanospheres and (NGPG/CS)_3_NPGP nanoparticles was also carried out.

### 2.6. Drug Loading and Release Research

DOX was used as a model anti-tumor drug to study the drug loading and controlled release properties of RGD-(NGPG/CS)_3_NPGP nanoparticles. The nanoparticles (10 mg) were dispersed in deionized water (2 mL), and then DOX aqueous solution (3 mg/mL, 1 mL) was added. The pH of the mixture was adjusted to about 6.0 and stirred for 24 h in the dark. After that, the pH of the solution was adjusted to 7.4 to put the nanoparticles in a contracted state and prevent premature drug leakage. The free DOX in the system was removed by centrifugation. The supernatant containing the free DOX was collected to calculate the free DOX content according to the established DOX standard curve and then obtain the drug loading (DLC) and drug encapsulation efficiency (DLE) according to the following formulas (1) and (2):(1)$$\mathrm{DLE}\;(\%)=\frac{\mathrm{weight}\;\mathrm{of}\;\mathrm{DOX}}{\mathrm{weigh}\;\mathrm{of}\;\mathrm{drug}-\mathrm{loaded}\;\mathrm{nanocapsules}}\times 100\%,$$
(2)$$\mathrm{DLC}\;(\%)=\frac{\mathrm{weight}\;\mathrm{of}\;\mathrm{DOX}}{\mathrm{weigh}\;\mathrm{of}\;\mathrm{DOX}\;\mathrm{in}\;\mathrm{feed}}\times 100\%.$$

The release behavior of DOX@RGD-(NGPG/CS)_3_NPGP nanoparticles in PBS solutions with different pHs was studied. Briefly, DOX@RGD-(NGPG/CS)_3_NPGP nanoparticles (10 mg) were dispersed in 2 mL of deionized water (stirred at 300 rpm for 15 min) and transferred to a dialysis tube (MWCO: 3.5 kDa) in a glass beaker containing 50 mL of dispersion (PBS solution with pH values of 5.0, 6.0, and 7.4). During the drug release experiment, at the set times, 3 mL of the release solution was gathered for testing, and buffers with the same pH and the same volume were added to the release system.

### 2.7. Cell Experiment

#### 2.7.1. Cytotoxicity Test

HCT116 cells and MCF-7 cells were cultured in DMEM with a high glucose content. The cytotoxicity of blank nanoparticles and DOX-loaded nanoparticles in the two cell lines was studied using the MTT assay. Cell viability was determined using the following equation:$$\mathrm{Cell}\;\mathrm{viability}\;(\%)=\frac{\mathrm{OD}\;\mathrm{sample}-\mathrm{OD}\;\mathrm{blank}}{\mathrm{OD}\;\mathrm{control}-\mathrm{OD}\;\mathrm{blank}}\times 100\%.$$

#### 2.7.2. Cellular Uptake Assays

HCT116 cells and MCF-7 cells were cultured at 0.5 × 10^6^ cells/well in 6-well plates and incubated with free DOX and DOX@RGD-(NGPG/CS)_3_NPGP with DOX at a concentration of 5.0 μg/mL for 4 h and 8 h. Then the cells were washed with PBS three times and fixed with a 4% paraformaldehyde solution at the predetermined time. The cell nuclei were stained with DAPI. The fluorescence images were captured by a laser scanning confocal microscope (CLSM).

## 3. Results and Discussion

### 3.1. Preparation and Characterization of Polyelectrolyte Multilayer-Coated SiO_2_(NGPG/CS)_3_NPGP Nanospheres

The schematic diagram of the preparation process of RGD-(NGPG/Cs)_3_NPGP nanoparticles is shown in Figure 1. Firstly, nanosized SiO_2_ was prepared by hydrolysis and condensation of TEOS in an ethyl orthosilicate (TEOS) water-base-alcohol system by the Stober method and then modified by APTES. Then, through the LbL assembly technology, the polysaccharides NPGP and chitosan were alternately wrapped on the template by electrostatic action. Finally, the nanoparticles are obtained by eroding the template.

Figure 2 shows the prepared nanosilica by the Stober method was regular, spherical in shape, uniform in size, and not agglomerated. The hydrated particle size of SiO_2_ nanospheres is 98.67 ± 6.33 nm by DLS. After modification by APTES, the synthesized SiO_2_-NH_2_ template is also spherical, while the size of the SiO_2_-NH_2_ template increases to 125.69 ± 16.42 nm and the zeta potential reaches 48.17 ± 0.61 mV, which is mainly due to the protonation of the amino group covered on the template surface and indicates that the modification of SiO_2_ nanospheres is successful.

SiO_2_(NGPG/CS)_3_NPGP nanospheres were prepared by using SiO_2_-NH_2_ as the template and driving the electrostatic interaction between the anionic NGPG polysaccharide and the cationic chitosan at an appropriate pH value using the layered assembly technique. The two polysaccharides attract each other through electrostatic interaction, and the molecular chains curl and tangle to achieve multilayer deposition [29]. At first, the SiO_2_-NH_3_ nanosphere templates had net positive charges. So, the initial step of polysaccharide multilayer construction was completed with the negatively charged NPGP solution. When the outermost layer is chitosan, some positive charges interact electrostatically with the NPGP in the inner layer, and the remaining net positive charges tend to extend the molecular chains of chitosan through electrostatic repulsion, and the mutual binding between molecular chains is loose. When the outer layer of chitosan is redeposited with NGPG, the situation is reversed. After the NGPG solution is added, there are two pathways for binding to SiO_2_(NPGP/CS)_X_ spheres. Most of the NPGP molecular chains are adsorbed to the outer surface of the SiO_2_(NPGP/CS)x sphere. Compared with the chitosan layer with strong electrostatic repulsion and a loose degree inside, the charge density of NPGP is lower, and the electrostatic repulsion between molecular chains is weak. In addition, there are many hydroxyl groups in the sugar ring, and the molecular chains are tightly fixed through the hydrogen bonding between hydroxyl groups. Due to the strong electrostatic repulsion between the molecular chains of the chitosan layer, the gap between the chains is large. A small part of the molecular chains will be inserted into the gap between the chitosan layers, combine with the protonated amino group on the chitosan on both sides, and compress the previously relaxed chitosan molecular layer. As shown in Figure 3b, the zeta potential of SiO_2_(NGPG/CS)_3_NPGP nanospheres changed alternately from positive to negative with the assembly layers increasing, indicating that the encapsulation of each layer was successfully achieved. In addition, the zeta potential of SiO_2_(NGPG/CS)_3_NPGP nanospheres is in the range of −30 mV to 30 mV, indicating that the nanoparticles are stable in the system and that the strong electrostatic repulsion between the particles hinders the aggregation of the particles.

The average particle size of SiO_2_(NGPG/CS)_3_NPGP nanospheres increased with polysaccharide deposition on the wall (Figure 3a). The hydrated average particle size ranged from 310.1 ± 8.564 nm to 429.73 ± 2.93 nm. TEM images also showed that the nanospheres were successfully prepared. As shown in Figure 3c, uniform spherical SiO_2_(NGPG/Cs)_3_NPGP nanospheres were observed. At high magnification (Figure 3d), the edge of the nanosphere with a thin layer is clearly observed, as indicated by the arrow, which could be the polysaccharide coating.

The chemical characteristics of SiO_2_, SiO_2_-NH_2_, NPGP, CS, and SiO_2_(NGPG/Cs)_3_NPGP were analyzed by FT-IR. As shown in Figure 4, the IR spectra of SiO_2_ before and after amino modification were basically consistent. The peaks near 1100 cm^−1^ and 800 cm^−1^ were the asymmetric contraction vibration peak and symmetric expansion vibration peak of Si-O-Si, respectively [30]. The absorption peak of SiO_2_ after modification at 1550 cm^−1^ was attributed to the symmetric bending vibration absorption peak of amino [31]. The results showed that APTES had been successfully grafted onto the surface of SiO_2_ nanospheres. In the IR of chitosan, the absorption peaks in the range of 3000 cm^−1^ to 3700 cm^−1^ are the characteristic peaks of -OH and -NH_2_. The peaks at 2867 cm^−1^ were the characteristic peaks of -CH_2_-, -CH_3_; the extension and contraction vibration of C=O (the non-deacetylated amide group on chitosan) was at 1650 cm^−1^; the characteristic peak of -NH_2_ was at about 1600 cm^−1^; and the absorption peaks at 1083 cm^−1^ and 1160 cm^−1^ were corresponding to C-O-C on the chitosan sugar rings [32]. For NPGP, 3400 cm^−1^ had a strong characteristic absorption band of -OH stretching vibration, 2950 cm^−1^ was the C-H stretching vibration absorption peak, 1017 cm^−1^ was the stretching vibration of cyclic ether C-O-C of polysaccharide, and the absorption peaks at 1736 cm^−1^ and 1606 cm^−1^ were the C=O stretching vibration peaks of methylated and unmethylated carboxyl groups, respectively. Compared with the IR of chitosan and NPGP, the absorption peak intensities of SiO_2_(NPGP/CS)_3_NPGP nanospheres in the range of 1600 cm^−1^–1700 cm^−1^ were weakened in the IR spectra, indicating that the electrostatic interaction between the amino group and ionized carboxyl group was the main driving force for the formation of nanospheres by the alternating precipitation of NPGP and chitosan on amino SiO_2_ [33].

### 3.2. Preparation and Characterization of Multilayer-Coated RGD-(NGPG/CS)_3_NPGP Nanoparticles

The inner SiO_2_-NH_2_ template was removed with an HF-NH_4_F buffer solution to obtain hollow nanoparticles. The reaction principle is as follows [34]:SiO_2_ + 4HF→SiF_4_ + 2H_2_O
SiF_4_ + 2HF→H_2_SiF_6_

The erosion rate of the SiO_2_-NH_2_ template depends on reaction time. As shown in Figure 5, the particle size of SiO_2_(NGPG/CS)_3_NPGP treated with HF-NH_4_F buffer solution decreased with the increase in reaction time. Within 1 h after the addition of HF-NH_4_F buffer, the particle size of the nanoparticles decreased significantly, which was due to the rapid disintegration of SiO_2_-NH_2_ template particles, which made the nanoparticles hard to support, so that the cyst wall contracted. After more than 1 h, the particle size was virtually unchanged with the increase in reaction time, and we can conclude that the template removal is almost complete and the nanoparticles are formed. In Figure 6, the overall hollow cavity and polysaccharide layer with a rough surface of the nanoparticles were clearly visible, but obvious black SiO_2_ particles could not be seen inside. The nanoparticles had certain shrinkage and adhesion but basically maintained their spherical shape.

Figure 7 is the TGA curve of SiO_2_(NGPG/CS)_3_NPGP nanospheres and (NGPG/CS)_3_NPGP hollow nanoparticles. Since SiO_2_ is a highly thermally stable compound with a melting point of 1723 ± 5 °C, the remaining 80% of SiO_2_(NGPG/Cs)_3_NPGP nanospheres heated to 800 °C from room temperature consist of SiO_2_ particles, and 20% of the weight loss is the weight of the CS-NPGP layers. By contrast, the weight loss rate of (NGPG/CS)_3_NPGP hollow nanoparticles was 80%, indicating that a small amount of SiO_2_ particles were not removed. The reaction time in the HF-NH_4_F buffer solution should be appropriately extended to completely remove the SiO_2_ particles.

By comparing the IR of SiO_2_(NGPG/CS)_3_NPGP nanospheres and (NGPG/CS)_3_NPGP hollow nanoparticles (Figure 8), it was found that the asymmetric contraction vibration peak of Si−O−Si was significantly weakened at 1095 cm^−1^, and the symmetric expansion vibration peak of Si-O-Si disappeared at 804 cm^−1^. All these changes indicated that the template SiO_2_−NH_2_ was successfully eroded, and the hollow nanoparticles were successfully built. The spectra after grafting with RGD peptide showed that the C=O stretching vibration absorption peak at 1637 nm^−1^ in RGD-(NGPG/CS)_3_NPGP and the new absorption peaks at 1520 nm^−1^ and 1253 nm^−1^ were the characteristic absorption peaks of N−H and the C−N stretching vibration peak of RGD peptide [35,36,37], indicating that the RGD peptide was successfully grafted onto the surface of nanoparticles.

### 3.3. Drug Loading and Release Behaviors

Compared with other drug carriers, hollow nanoparticles have attracted much attention due to their high loading of small-molecule drugs and protein macromolecules. After the inclusion of doxorubicin, the zeta potential of RGD-(NGPG/CS)_3_NPGP nanoparticles increased from −37.23 ± 1.11 mV to −22.18 ± 2.13 mV, which might be due to the partial adsorption of positively charged DOX molecules on the surface of the nanoparticles by static electricity. In addition, the size of the drug-loaded nanoparticles increased from 345.20 ± 1.58 nm to 407.23 ± 2.54 nm, compared with that of the non-drug-loaded nanoparticles. The entrapment efficiency (EE) and loading capacity (DLC) of the prepared RGD-(NPGP/CS)3NPGP nanoparticles for DOX were 83.23 ± 6.12% and 76.51 ± 1.24%, respectively. The study showed nanoencapsulation using pectin as a wall material with high encapsulation efficiency and controlled delivery of bioactive compounds. For example, the successfully encapsulated polyphenol oleuropein in olive leaf with pectin achieved an encapsulation efficiency of 91% under optimum conditions. A study on nanoencapsulation of vitamin E used the method of double emulsion using pectin and k-carrageenan complexes, which had an encapsulation efficiency of 66%. Thus, the use of pectin for nanoencapsulation based on special properties can be helpful in controlled, targeted delivery of bioactive compounds in different conditions and increased bioaccessibility [38].

Among various stimulation-responsive controlled release systems, pH-responsive drug delivery systems have a broad application prospect because the tumor microenvironment exhibits low pH, especially intracellular bodies (pH = 5.0–6.0) and lysosomes (pH = 4.5–5.0) [39]. In this study, the release behavior of DOX-loaded RGD-(NGPG/CS)_3_NPGP nanoparticles in a release medium with pH at 7.4, 6.0, and 5.0 was investigated individually. As shown in Figure 9, DOX exhibited multiple release behaviors at different pHs, and the release rate increased with the decrease in pH. In 75 h, only approximately 23.22% of DOX was released in a physiological microenvironment with a pH of 7.4. The total drug release in the medium at pH 6.0 was about 53.73%. At pH 5.0, the drug release within 10 h was 40.71%, and the cumulative release within 48 h was 74.41%. Therefore, RGD-(NGPG/CS)_3_NPGP nanoparticles showed pH-responsive properties. It should be noted that in the process of release, due to the time difference between the diffusion of DOX from inside the dialysis bag to outside the band, there is also a small amount of free DOX lost in the dialysis bag, which leads to the loss of part of the DOX in the process of release. Due to the dilution effect of the dialysis system, the actual release amount is slightly larger than that measured, but the overall trend remains unchanged.

As a low-methoxyl pectin, the electrostatic repulsion between NPGP molecules gradually decreased with a decrease in pH. In natural conditions, NPGP had a high charge density, showed high intermolecular electrostatic repulsion, and prevented the aggregation of NPGP molecules. With the pH decreasing continuously, the zeta potential was significantly decreased, and the electrostatic repulsion between NPGP polysaccharides was greatly attenuated, which promoted the formation of intermolecular hydrogen bonds and ultimately led to gelation [20]. The pH-responsive property is considerably enhanced due to the strengthened hydrogen bonding between NPGP molecules and then the suppressed binding between the amino groups of chitosan and the carboxyl groups of NGPG in different layers at low pH. In detail, owing to the weak polyelectrolyte properties of CS and NGPG, their charge densities and conformations could be regulated by changing the pH value. For chitosan (pKa, 6.5), when pH < 6.5, the positive charge density increased due to the protonation of the excess free amino groups, which would lead to the nanoparticles in a swelling state and further induce the increase in permeability. In addition, the repulsive forces between DOX·HCl and the protonated amino groups in chitosan were also increased. These performances facilitated the release of the drugs both in the cavities and in the interlayer via free diffusion at a lower pH. For NGPG (pKa 3.5), the negative charge density decreased under low pH microenvironments owing to the deionization of the ionized carboxyl groups, which could decrease the electrostatic attractions between NGPG and CS and result in the formation of a looser layer to accelerate the drug release. Free adriamycin has good water solubility, a small molecular weight, and can be released quickly through a dialysis bag at different pH values. Therefore, our nanoparticles have a sustained release effect on DOX in an acidic microenvironment and can prevent DOX release in a physiological microenvironment.

### 3.4. Cytotoxicity Study

First, the in vitro cytotoxicity of RGD-(NGPG/CS)_3_NPGP nanoparticles on HCT-116 cells and MCF-7 cells was studied. The cells were treated with RGD-(NGPG/CS)_3_NPGP nanoparticles at different concentrations for 48 h. As shown in Figure 10, even when treated with the blank nanoparticles at 800 μg/mL, the cell viability of HCT-116 cells and MCF-7 cells was above 80%, indicating that the hollow nanoparticles had no cytotoxicity. In addition, the anti-tumor activity of DOX@RGD-(NGPG/CS)_3_NPGP nanoparticles on HCT-116 and MCF-7 cells in vitro was detected by MTT assay. As the literature reports, the integrin αvβ3 is highly expressed in HCT-116 cells, while it is almost not expressed in MCF-7 cells [40].

Figure 11 shows the concentration-dependent killing effects of free DOX and DOX@RGD-(NGPG/CS)_3_NPGP nanoparticles on two cancer cell lines. In theory, free DOX is more likely to enter the cell and exert a cytotoxic effect than nonencapsulated DOX. While the free DOX and DOX@RGD-(NGPG/CS)_3_NPGP nanoparticles showed similar cytotoxic effects on HCT-116 cells. However, for MCF-7 cells, the cytotoxicity of DOX@RGD-(NGPG/CS)_3_NPGP nanoparticles was significantly lower than that of free DOX, especially when the DOX concentration was higher than 5 μg/mL.

This might be because the nanoparticles grafted with RGD peptide could actively recognize the overexpression of αvβ3 receptor on the surface of tumor cells in HCT-116 cells, thus increasing phagocytosis and uptake efficiency and then promoting intracellular release. The integrin αvβ3 is almost not expressed in MCF-7 cells. The DOX@RGD-(NGPG/CS)_3_NPGP is hard to touch with the surface of the cells through the recognition between the RGD peptide and integrin αvβ3. Thus, the cytotoxicity of DOX@RGD-(NGPG/CS)_3_NPGP nanoparticles was lower than that of free DOX. These comparisons indicated that RGD peptide modification of the DOX@RGD-(NGPG/CS)_3_NPGP helped to identify targeted cells and promote further endocytosis.

### 3.5. Cellular Uptake of DOX-Loaded Nanoparticles

The uptake of RGD-(NGPG/CS)_3_NPGP nanoparticles by tumor cells HCT-116 and MCF-7 was observed by CLSM. HCT-116 cells showed high expression of the integrin receptor on the surface, while MCF-7 cells showed low expression. Therefore, these two cells were used to further investigate the targeting of nanoparticles. In Figure 12, blue represents the nucleus, red represents DOX, and merged is the overlaid picture of the former two.

For HCT-116 cells, the red fluorescence appeared in the nucleus after the nanoparticles were incubated with the cells for 4 h, indicating that the DOX-loaded carrier was effectively internalized and released into the nucleus. In addition, the nuclei in the free DOX group also showed obvious red fluorescence. While for MCF-7 cells, after the same incubation for 4 h, the RGD-(NGPG/CS)_3_NPGP group only had weak red fluorescence. After incubation for 8 h, HCT-116 cells ingested more nanoparticles, showing a higher red fluorescence intensity. Although the uptake by MCF-7 cells increased after 8 h of incubation, it was much smaller than that of HCT-116 cells. Therefore, RGD-(NGPG/CS)_3_NPGP nanoparticles could specifically recognize HCT-116 cells with high expression of integrin αvβ3, and improve the internalization and phagocytosis of cells, leading to an increase in DOX accumulation in cells. The uptake experiment also proved the targeting of RGD-(NGPG/CS)_3_NPGP nanoparticles.

## 4. Conclusions

In this study, based on excellent gel properties and pH response, a new pectin NPGP with a high galactouronic acid and low methoxylation degree was chosen to couple with chitosan as capsule wall materials and SiO_2_ as a template to prepare a novel nanoparticle with pH response and integrin targeting for the delivery of anticancer drugs by layer-by-layer assembly technology. The entrapment efficiency (EE) and loading capacity (DLC) of the prepared RGD-(NPGP/CS)_3_NPGP nanoparticles for DOX were 83.23 ± 6.12% and 76.51 ± 1.24%, respectively. In addition, RGD-(NGPG/CS)_3_NPGP nanoparticles can be easily absorbed by HCT-116 cells with high expression of integrin αvβ3, and rapidly release DOX under the stimulation of low intracellular pH, thereby exerting anti-tumor effects.

Therefore, the NPGP-chitosan nanoparticles are good candidates as a drug delivery system. This work is expected to open a new strategy for the design of drug carriers based on natural polysaccharides. The stability, safety, further targeting, and efficacy of the nanoparticles at the animal level need further experimental studies.

## Figures and Tables

**Figure 1 polymers-15-02510-f001:**
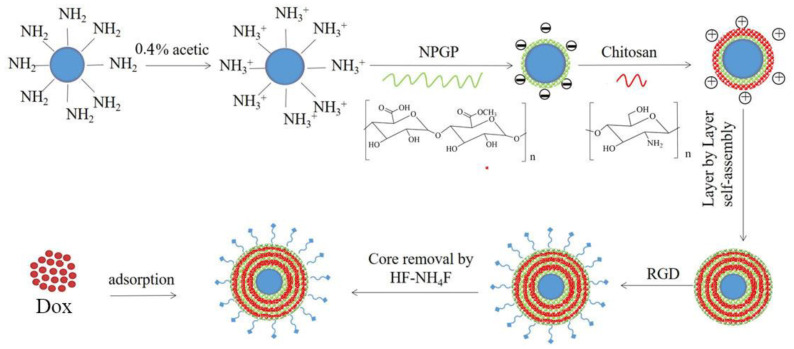
Schematic diagram of the formation process of NPGP nanoparticles via LbL self-assembly.

**Figure 2 polymers-15-02510-f002:**
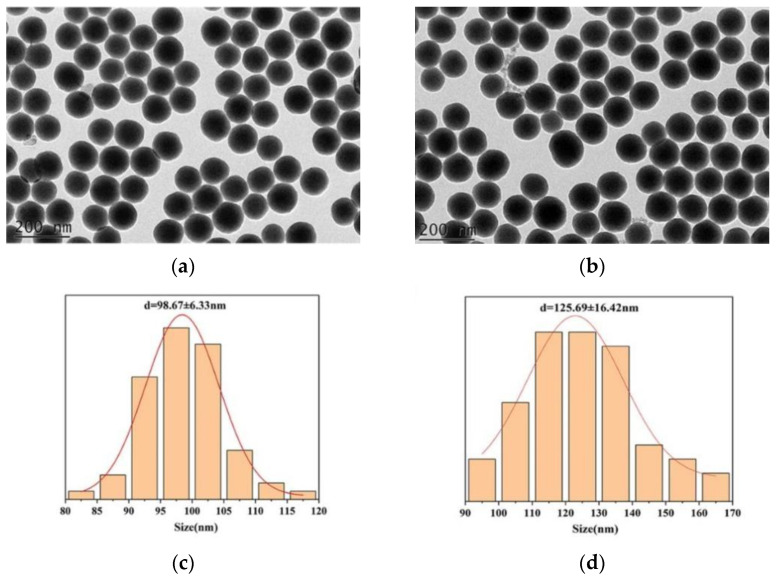
TEM images of SiO_2_ (**a**) and SiO_2_-NH_2_ (**b**). Size of SiO_2_ (**c**) and SiO_2_-NH_2_ (**d**).

**Figure 3 polymers-15-02510-f003:**
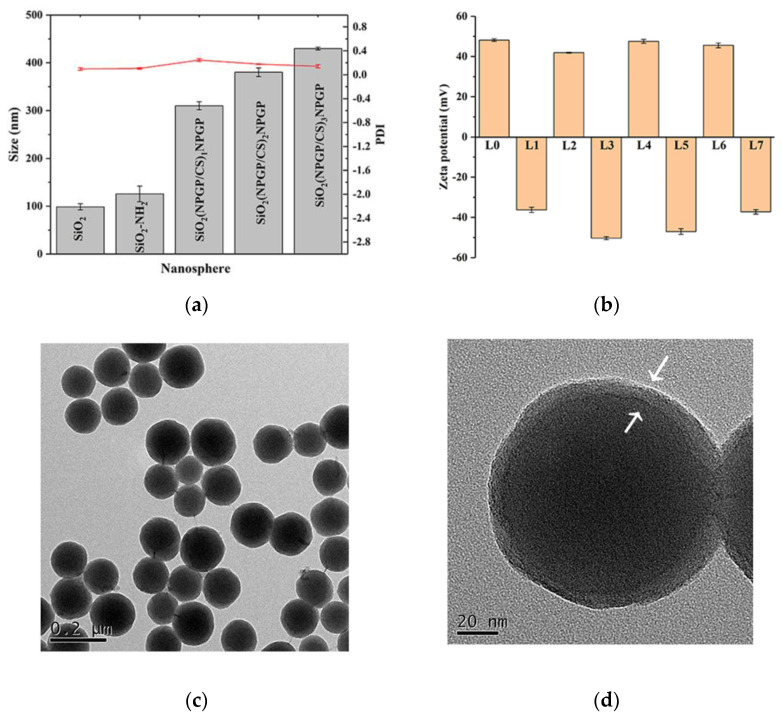
Size, PDI, zeta potential, and morphology of the series of nanospheres during the preparation of SiO_2_(NGPG/CS)_3_NPGP nanospheres. (**a**) The average hydrodynamic diameter, PDI (The bar chart means size, the red line means PDI), and (**b**) the zeta potential of nanospheres in deionized water with different deposited layers. TEM images of SiO_2_(NGPG/CS)_3_NPGP at low (**c**) and high magnification (The arrows point out the polysaccharides layer) (**d**).

**Figure 4 polymers-15-02510-f004:**
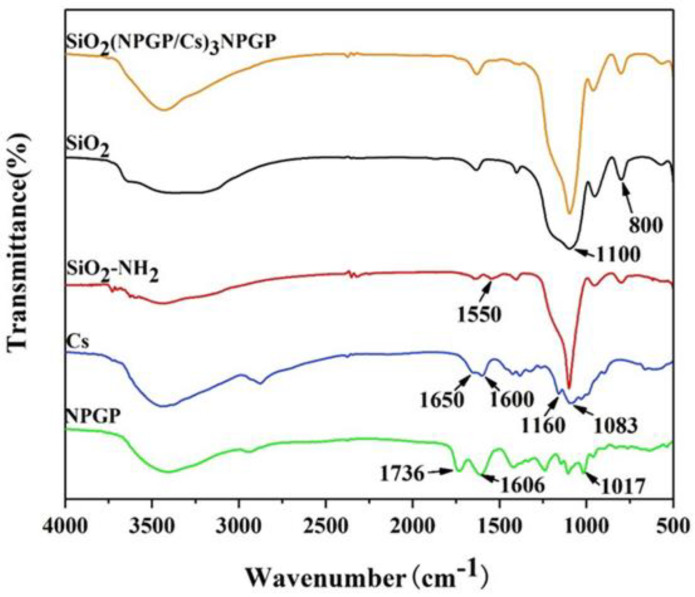
FTIR spectra of SiO_2_ nanospheres, SiO_2_-NH_2_ nanospheres, CS, NPGP, and polyelectrolyte multilayer-coated SiO_2_(NGPG/CS)_3_NPGP nanospheres.

**Figure 5 polymers-15-02510-f005:**
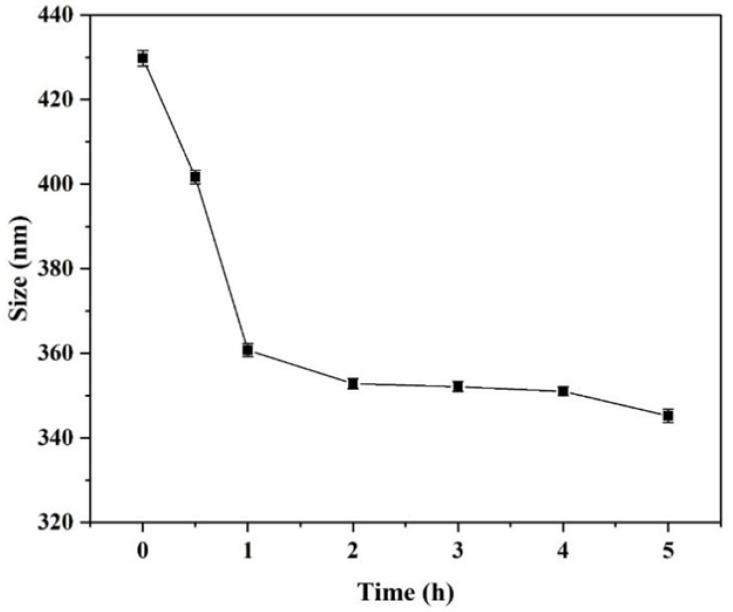
The trend of the average hydrodynamic diameter of polyelectrolyte multilayer-coated SiO_2_(NGPG/CS)_3_NPGP nanospheres exposed to HF-NH_4_F solution.

**Figure 6 polymers-15-02510-f006:**
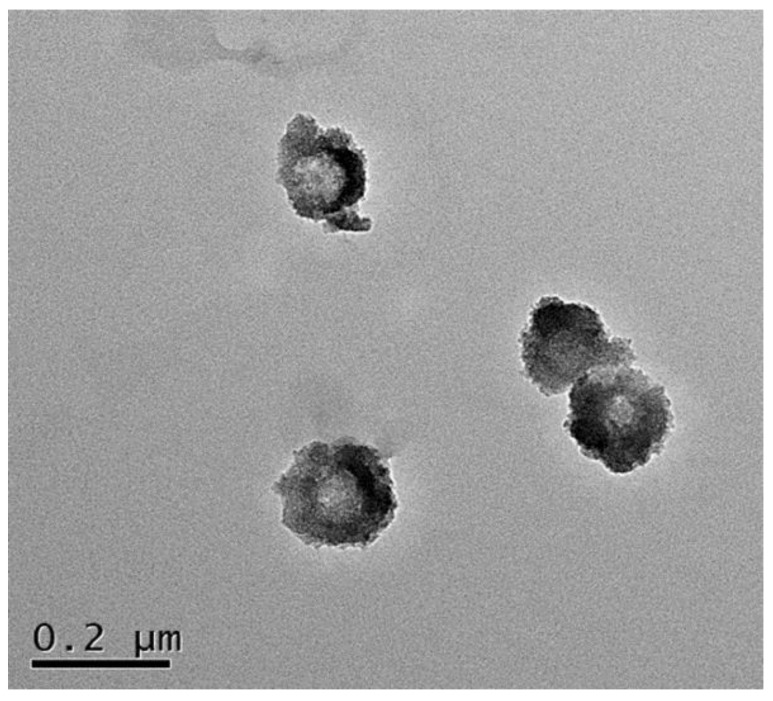
TEM images of polyelectrolyte multilayer-coated (NGPG/CS)_3_NPGP nanoparticles.

**Figure 7 polymers-15-02510-f007:**
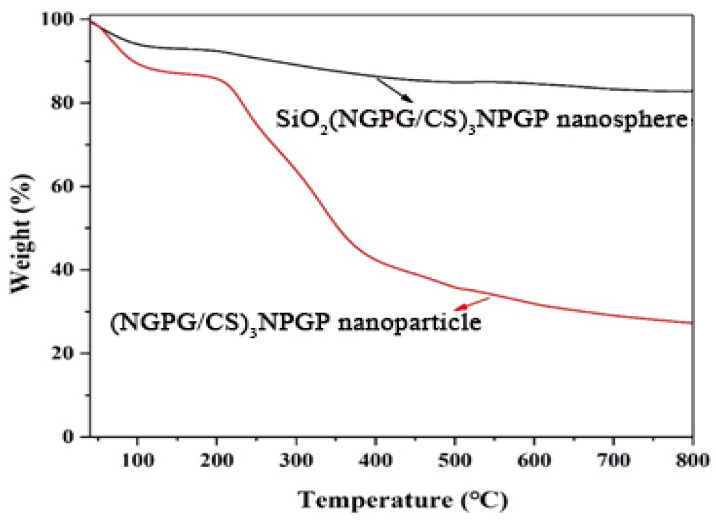
TGA thermograms of SiO_2_(NGPG/CS)_3_NPGP nanospheres and (NGPG/CS)_3_NPGP nanoparticles.

**Figure 8 polymers-15-02510-f008:**
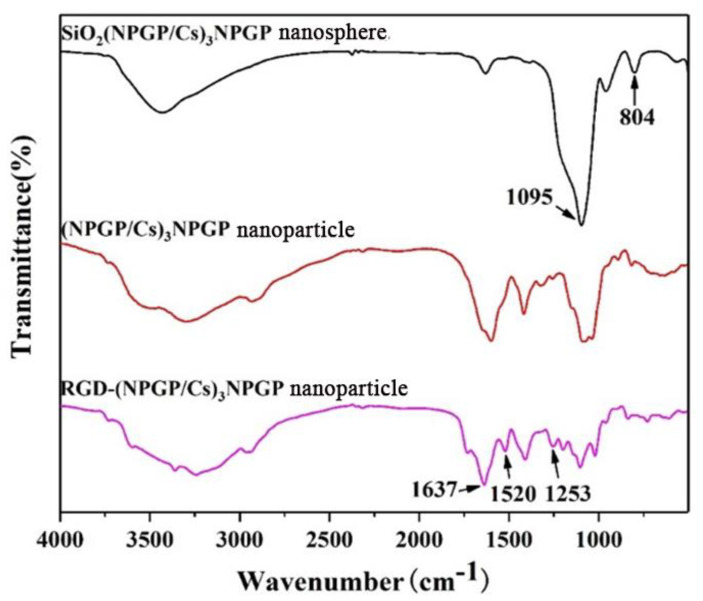
FTIR spectra of SiO_2_(NGPG/CS)_3_NPGP nanospheres, (NGPG/Cs)_3_NPGP nanoparticles, and RGD-(NGPG/CS)_3_NPGP nanoparticles.

**Figure 9 polymers-15-02510-f009:**
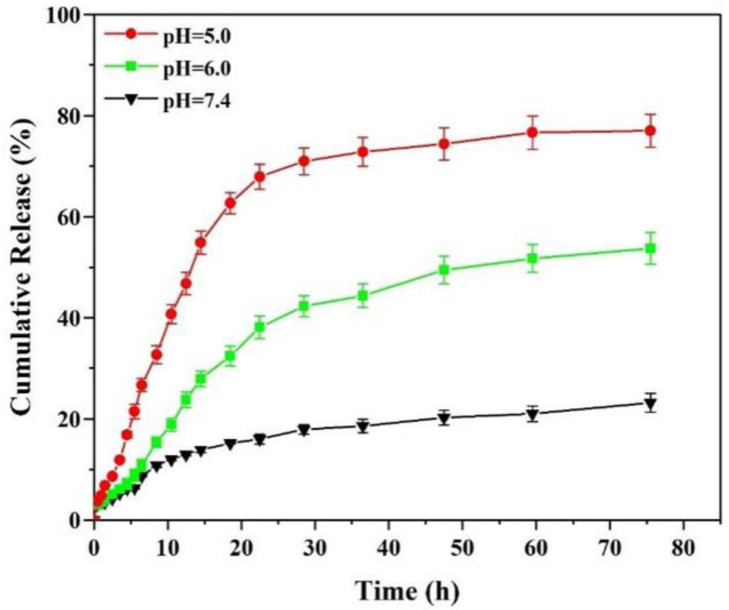
Nanoparticle cumulative release of DOX from DOX@RGD-(NGPG/CS)_3_NPGP nanoparticles.

**Figure 10 polymers-15-02510-f010:**
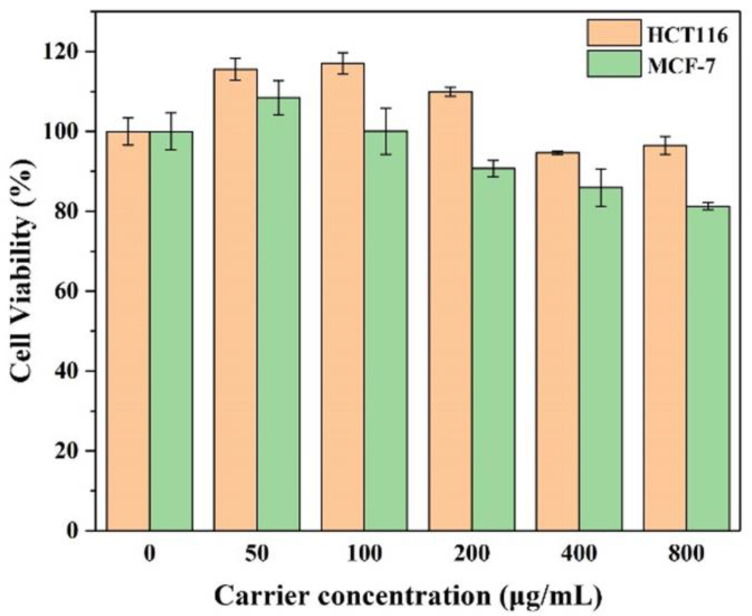
Cell viability of HCT116 and MCF-7 cells after 48 h of incubation with nanoparticles at different concentrations (the data was expressed as mean ± SD, n = 3).

**Figure 11 polymers-15-02510-f011:**
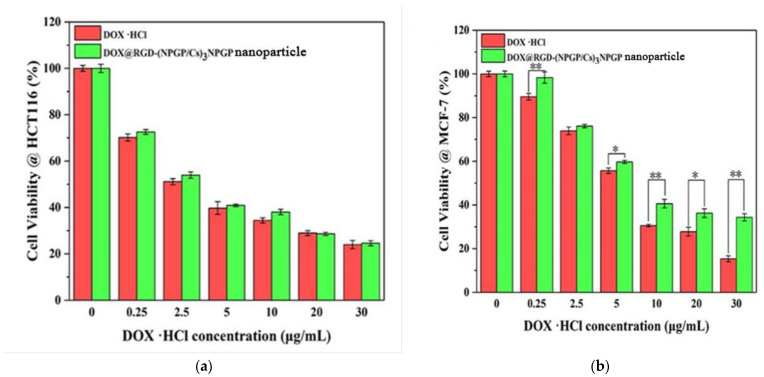
Anticancer activity of free DOX and DOX@RGD-(NPGP/CS)_3_NPGP nanoparticles against HCT-116 cells (**a**) as a function of DOX concentration after 48 h incubation; anticancer activity of free DOX and DOX@RGD-(NPGP/Cs)_3_NPGP nanoparticles against MCF-7 cells (**b**) as a function of DOX concentration after 48 h incubation. (** *p* < 0.01, * *p* < 0.01).

**Figure 12 polymers-15-02510-f012:**
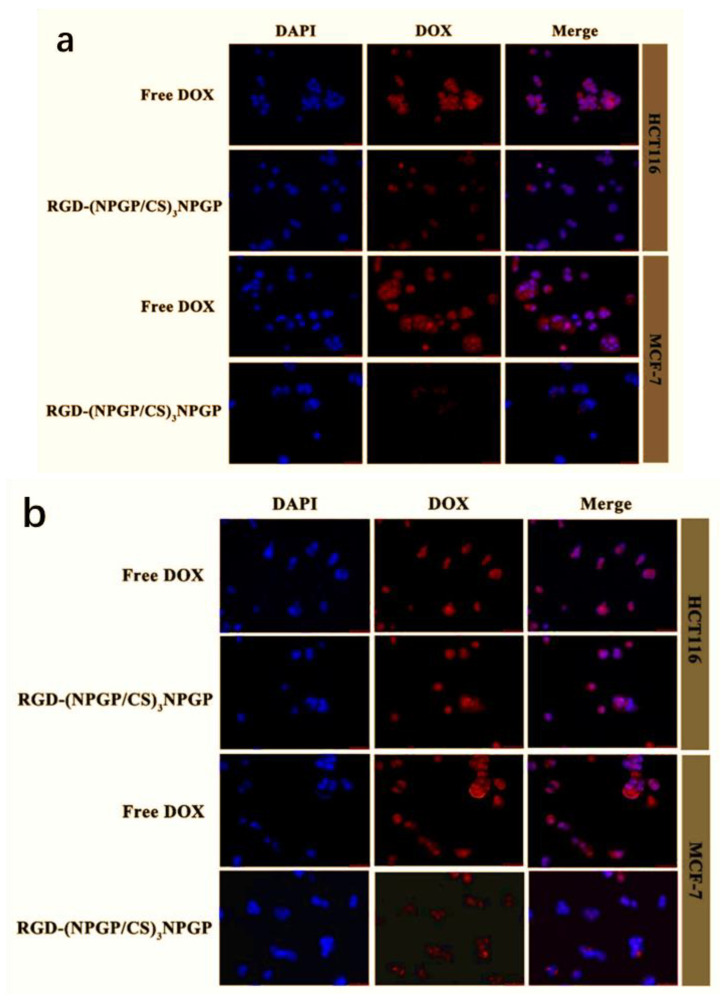
CLSM images showing intracellular uptake of DOX and DOX@RGD-(NGPG/Cs)3NPGP nanoparticles by HCT-116 and MCF-7 cells after 4 h (**a**) and 8 h (**b**).

## Data Availability

Data is unavailable due to privacy or ethical restrictions.

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
