# Peer review of "Preparation and Properties of Natural Polysaccharide-Based Drug Delivery Nanoparticles"

_polymers, 2023, doi:10.3390/polym15112510_

Round 1
Reviewer 1 Report
The manuscript covers the use of NPGP for the preparation of nanocapsules via layer-by-layer coating of sacrifice SiO2. Hollow particles were further modified by RGD peptide grafting, and in-vitro cell targeting was evaluated.
Some comments are listed below:
1) The authors mention that the extraction and characterization on NPGP were assessed in previous experiments, but no references were included in the Introduction or Materials Sections. Relevant references should could contribute to the reader’s comprehension about the main “new” raw material (source and physicochemical properties). A brief sentence can be added to summarize the main features.
2) The yield of the SiO2 removal process is not clear. The authors conclude that the template removal is complete because of size remain unchanged (though a small decrease can be observed at 5h) and particles could not be seen inside. However, they later affirm that “a small amount of SiO2 particles were not removed” and further reaction time is required. Further on, it is again stated that “template SiO2-NH2 is successfully eroded” because of the Si-O-Si peak weakening (no disappearing) by FTIR. Statements should be consistent throughout the manuscript, and the discussions must be supported by all the tests carried out with different techniques.
3) FTIR: How do you explain the new large peaks in (NGPG/Cs)3NPGP hollow nanocapsules (when compared with SiO2(NGPG/CS)3NPGP nanospheres)? Were there unexpected reactions/processes derived from removal treatment (HF-NH4F)?
4) The authors describe the effect of a continuous pH decrease on the behavior of NPGP molecules and their Z potential. It may be interesting to show these results. From the protocol it is assumed that different buffers were used for the release tests at different pHs (line 272), but not gradual pH-changes. Under what circumstances do you expect gelation in this system? How this would affect the nanocapsules behavior? Please hypothesize how the decrease in pH (formation of intermolecular hydrogen, etc.) caused an increase in the release rate.
5) Please include some missing details on the experimental protocols. For example, how was centrifugation (line 128) carried out (time, speed, did it somehow damage the capsules?, etc.). How was DOX analytically determined (e.g. UV)?
Reviewer 2 Report
|
Round 2
Reviewer 1 Report
The authors adequately addressed all comments and suggestions in the new version. New references, experimental details and further discussion were incorporated. Thank you.
Author Response
Thank you for your review and approval.
Reviewer 2 Report
Preparation and characterization of layer-by-layer (LbL) assembly nanoparticles using biopolymers is hard to achieve. Data provided do not demonstrate that LbL assembly nanocapsules were obtained and accordingly this achievement cannot be claimed by authors and its mention as an achievement cannot be found in the manuscript. In the same way the term “nanocapsules” cannot be claimed either. Authors must provide data or discuss obtained results within this scenario,i.e. it is all about ionic interactions as there are plenty of published nanostructures (other than LbL) based on these interactions.
1.. Full and detailed characterization of biopolymers including size (Molecular weight of “Nicandra physalodes” (Linn.) Gaertn (NPGP) and chitosan are not identical contrarily to what the graphical abstract suggest), charge as well of doxorubicin (DOX). What does DOX is encapsulated (does it interact with both polysaccharides?), and what happens during release studies;
2. The meaning of drug loading (DL) and drug encapsulation efficiency (LCE) is doubtful as no clear comparison of them with literature data can be made;
3. There is no evidence of hollow core-shell structures. TEM images provided (Figure 3d and Figure 6) are far from enough to make clear the distinction, and zeta potential oscillating values can be a meaning of structure unspecific conjugates.
4. There are three main reasons to perform release studies with free DOX. First, it plays like a control for DOX quantification. Second, assessing the role on DOX release is useful, particularly when burst and delayed release are likely to occur. Third and not less important, how stable is DOX in release media? Authors should point out these limitations of their work.
5. Please discriminate the amount of DOX@RGD-(NGPG/Cs)3NPGP nanocapsules dispersed in 2 mL of deionized water before transferring to release medium and the modality of agitation (including speed) and rationale for choosing a 3,5 kDa MWCO dialysis bag.
6.The selection of an optimal technique for sample preparation is the most important step in setting up an analytical method. -Please provide UV quantification is DOX including volume of withdrawn samples and processing before reading absorbance; Provide also final pH of chitosan and NPGP solutions.
Round 3
Reviewer 2 Report
1.Data provided do not demonstrate that LbL assembly nanocapsules were obtained, and accordingly, this achievement cannot be claimed by authors and its mention as an achievement cannot be found in the manuscript. In the same way, the term “nanocapsules” cannot be claimed either. The polyelectrolyte multilayer-coated SiO2-NH2 nanospheres fit the accurate term until authors gather more evidence of LbL nanocapsules formation. Discussion regarding polyelectrolytes interactions has improved but lack of supporting bibliographic references.
2.Schematic representation of polyelectrolyte multilayer-coated SiO2-NH2 nanospheres is fair even tough, and according to authors “Nicandra physalodes” (Linn.) Gaertn (NPGP) layer will not have the thickness of chitosan one. Comparison to Figure from Ji, F.; Li, J.; Qin, Z.; Yang, B.; Zhang, E.; Dong, D.; Wang, J.; Wen, Y.; Tian, L.; Yao, F. Engineering pectin-based hollow nanocapsules for delivery of anticancer drug. Carbohydrate Polymers 2017, 177, 86-96, doi:https://doi.org/10.1016/j.carbpol.2017.08.107 can be made but with caution as pectin and chitosan used are not the same.
3.The meaning of drug loading (DL) and drug encapsulation efficiency (LCE) is still unclear. Authors must check the use of Ng in the formula (is it NP?). How do authors compare the amount of drug encapsulated used with the ones available in literature? How does the technique work well regarding the amount of doxorubicin required for …biological assays….and amount of initial drug used for encapsulation (is this the “weight of doxorubicin in the feed”)?
4. There is no evidence of hollow core-shell structures. TEM images provided (Figure 3d and Figure 6) are far from enough to make clear the distinction, and zeta potential oscillating values can be a meaning of structure unspecific conjugates.
5. Fig. 6 shows the TEM photographs of the polysaccharide nanospheres of SiO2(NPGP/CS)3NPGP. It is impossible to follow the authors when they state, “.. polysaccharide nanocapsules have a hollow cavity and thin polysaccharide wall with a rough surface”.
6. Authors are asked to state that some doxorubicin could have been lost during release experiments. Rather than decreasing the input brought by their work, this remark increases rigor and accuracy instead.
6. The selection of an optimal technique for sample preparation is the most important step in setting up an analytical method. -Please provide UV spectra of both doxorubicin and unloaded nanospheres. Please make clear the following statement “and the mother solution was diluted into DOX solution (1, 2, 4, 8, 16 ug/mL)”. Authors are still not provided with the volume of withdrawn samples and replacing withdrawn medium procedure, if any.
Round 4
Reviewer 2 Report
Data provided to demonstrate that NPGP/chitosan coated nanospheres were obtained, and this achievement can be claimed by authors but the mention of LbL assembly nanocapsules cannot be found in the manuscript. In the same way, the term “nanocapsules” must be replaced by nanospheres (nanoparticles), nanoconjugates or polyelectrolyte nanostructures. Moreover, the term “shell” must be replaced by “layer”. There is no evidence of a core-shell structure so authors must revise the terminology used in the manuscript. As suggested, polyelectrolyte multilayer-coated SiO2-NH2 nanospheres fit the accurate term until authors gather more evidence of LbL nanocapsules formation.
Author Response
Thank you for your strict and meticulous review which is useful for our research and experimental design in the future. We accepted your suggestion and revised the expression in the article and figure.
